# Natural hazard impacts on transport infrastructure in Russia

Elena Petrova

Faculty of Geography, Lomonosov Moscow State University, Moscow, 119991, Russia

*Correspondence to*: Elena Petrova (epgeo@mail.ru)

**Abstract.** The transport infrastructure of Russia is exposed to multiple impacts of various natural hazards and adverse weather phenomena such as heavy rains and snowfalls, river floods, earthquakes, volcanic eruptions, landslides, debris flows, snow avalanches, rock falls, ice phenomena, and others. The paper considers impacts of hazardous natural processes and phenomena on transport within the area of Russia. Using the information of the author's database, contributions of natural factors to road, railway, air, and water transport accidents and failures are

assessed. The total risk of transport accidents and traffic disruptions triggered by adverse and hazardous natural impacts, as well as the risk of road and railway accidents and disruptions as the most popular modes of transport is assessed at the level of Russian federal regions. The concept of emergency situation is used to measuring risk. In the risk analysis, 838 emergency situations of various scale and severity caused by natural hazard impacts on the transport infrastructure over 1992 to 2018 are considered. The average annual number of emergencies is taken as an

indicator of risk. Regional differences in the risk of transport accidents and disruptions due to natural events are analyzed. Regions most at risk are identified.

**Keywords**: Transport infrastructure, natural hazards, transport accident, traffic disruption, database

## 1 Introduction

According to the Federal Law "On Transport Security" (2019), transport infrastructure of the Russian Federation

(RF) is considered as a large and complex technological system including tunnels, overpasses, and bridges; terminals and stations; river and sea ports; airports; roads, railways, and waterways, as well as other buildings, structures, and equipment ensuring the functioning of the transport system. Russia has a very extensive transportation network that is among the largest in the world. It includes 1.5 million km of public roads, more than 600,000 km of airways, 123,000 km of railway tracks, and 100,000 km of inland navigable waterways (Rosstat,

25 2018).

In studies on the impacts of natural hazards, transport infrastructure is most often classified by mode of transport including road, rail, water, and air transport (e.g. Govorushko 2012; Mattsson and Jenelius, 2015; Voumard et al., 2018). Some researchers classify it by infrastructure asset types. For example, Kaundinya et al. (2016) select such transport assets as bridges, tunnels, embankments, cuts, and centralized systems. This analysis is structured by mode

of transport.

Due to the large length of the transportation network, as well as climatic, geological, geomorphologic, and other natural features of the country, transport infrastructure facilities of Russia are exposed to the undesirable impacts of adverse natural processes and phenomena, as well as natural hazards of various genesis, such as geophysical, hydro-meteorological, and others. Distribution of various natural hazards through the country area is discussed below in

Sect. 2.1. Their impacts may endanger transport safety and reliability, trigger accidents and failures, disrupt the normal operation of transport system, cause delays in delivery of passengers and goods, and lead to other negative consequences.

Natural processes and phenomena can be classified in various ways depending on the objectives of a study. Natural hazards can be typify according to their genetic features (e.g. Voumard et al., 2018), the intensity of their manifestation, the main formation and development factors, characteristics of spatial distribution and mode, etc. (Malkhazova and Chalov, 2004). Liu et al. (2016) propose a systematic natural hazard interaction classification based on the hazard-forming environment. Gill and Malamud (2016) propose a detailed classification of natural-hazard types in Guatemala, including six natural-hazard groups (geophysical, hydrological, shallow Earth processes, atmospheric, biophysical, and space), 19 hazard types, and 37 hazard sub-types.

Previously, two types of natural hazards were found by the author, based on their genesis, distribution in space and time, and the impact pattern on the technosphere and society in populated areas (Petrova, 2005). In the context of the present study, the proposed classification scheme was adapted taking into account impacts of natural hazards on the transport infrastructure (Fig. 1).

Solar and geomagnetic disturbances (space weather), geodynamics, geophysical and astrophysical field variations, and other global processes belong to the first group. They have global scale in space and cyclic development in time. Natural processes of this type may influence the transport infrastructure both directly, causing electronics error and automatic machinery failure, as well as indirectly, by affecting the nervous system of operators, drivers or pilots and thereby leading to a decrease in their reliability. Natural hazards of the second type are of more "earthly" origin, i.e. from the atmosphere, lithosphere, hydrosphere or biosphere. They vary greatly in their spatial scale and geographical location. This type of natural hazards includes earthquakes, volcanic eruptions, landslides, snow avalanches, hurricanes, windstorms, heavy rains, hail, lightning, snow and ice storms, temperature extremes, wild fires, floods, droughts, etc. Natural hazards belonging to this group cause a direct destructive effect leading to accidents and disruptions.

A transport accident is any accident that occurs when people and goods are transported. With over 1.2 million people killed each year, road accidents are among the world's leading causes of death; another 20–50 million people are injured each year on the world's roads (WHO, 2017). Transport accidents of other types including air, rail, and water transport are not as numerous as road crashes, but the severity of their consequences is much higher because of the higher number of people killed and injured per accident. Shipwrecks with a large number of passengers have the highest number of casualties.

Traffic interruptions and disruptions cause multiple social problems because our societies are highly dependent on the transport system for people's daily mobility and for goods transport (Mattsson and Jenelius, 2015). In the case of emergency situation, transport network serves as a life-line system. Thus, ensuring the robustness and reliability of the transport system is one of the most important and pressing problems of the socio-economic development of any country. In May 2018, the Ministry of Transport of the RF has developed a new version of the Transport strategy up to 2030 (Ministry of Transport of the Russian Federation, 2018). Among the key priorities, the Transport strategy includes requirements to cope with the modern challenges, such as climate change and a need for increasing the safety of the transport system.

Since the early 1950's (Tanner 1952), it has been recognized that weather conditions affect many road (un-)safety aspects such as driver's attention and behavior, vehicle's operation, road surface condition, etc. A large number of studies devoted to the influence of weather factors on the accident rates were published over the last decades. All the authors agree that the adverse weather is a major factor affecting road situation (e.g. Edwards 1996; Rakha et al 2007; Andrey 2010; Andersson and Chapman 2011; Bergel-Hayat et al 2013; Chakrabarty and Gupta 2013). Many authors connect the maximum number of road accidents with precipitations (Jaroszweski and McNamara 2014; Spasova and Dimitrov 2015). Aron et al (2007) revealed that 14 % of all injury accidents in Normandy (France)

took place during rainy weather and 1 % during fog, frost or snow/hail. Satterthwaite (1976) found the rainy weather to be a major factor affecting accident numbers on the State Highways of California: on very wet days the number of accidents was often double comparing to dry days. Brodsky & Hakkert (1988) with data from Israel and the USA did indicate that the added risk of an injury accident in rainy conditions can be two to three times greater than in dry weather; when a rain follows a dry spell, the hazard could be even greater. Among other weather factors, bright sunlight was identified as a cause of accidents (Shiryaeva 2016). Redelmeier and Raza (2017) investigated visual illusions created by bright sunlight that lead to driver error, including fallible distance judgment from aerial perspective. According to their results, the risk of a life-threatening crash was 16 % higher during bright sunlight than normal weather.

Some authors consider other natural hazards, such as landslides (Bíl et al., 2014; Schlögl et al., 2019), flash floods (Shabou et al., 2017) or rock falls (Bunce et al., 1997; Budetta and Nappi, 2013).

As for railway transport, most of papers also focus on specific hazards, considering impacts of adverse weather and hydro-meteorological extremes (Ludvigsen and Klæboe, 2014; Nogal et al., 2016), landsliding (Jaiswal et al., 2011), flooding (Hong et al., 2015; Kellermann et al., 2016), snowfall (Ludvigsen and Klæboe, 2014) or tree falls (Nyberg and Johansson, 2013; Bil et al., 2017) as triggers of accidents.

Some studies combine all types of natural hazards affecting road and rail infrastructure (Govorushko 2012; Petrova, 2015; Kaundinya et al., 2016). Voumard et al. (2018) examine small events like earth flow, debris flow, rock fall, flood, snow avalanche, and others, which represent three-quarters of the total direct costs of all natural hazard impacts on Swiss roads and railways.

Investigations of natural hazard impacts on other transport systems than roads and railways are not so numerous. As example, studies about danger of volcanic eruptions to the aviation should be mentioned (Neal et al, 2009; Brenot et al., 2014; Girina et al., 2019). Large explosive eruptions of volcanoes can eject several cubic kilometers of volcanic ash and aerosol into the atmosphere and stratosphere during a few hours or days posing a threat to modern airliners (Gordeev and Girina, 2014).

Only few researches investigate impacts of global processes, such as geomagnetic storms (space weather) and seismic activity. In the early 1990's, Epov (1994) found a correlation (R=0.74) between solar activity and temporal distribution of air crashes. Desiatov et al. (1972) argue that the number of road accidents multiplies by four on the second day after a solar flare in comparison to "inactive" solar days. According to Miagkov (1995), solar activity affects operators, drivers, pilots, etc., causing a "human error" and "human factor" of accidents. Kanonidi et al. (2002) study a relationship between disturbances of the geomagnetic field and the failure of automatic railway machinery. Kishcha et al. (1999), Anan'in and Merzlyi (2002) examine a correlation between seismic activity and air crashes.

The main purpose of this study is to investigate impacts of natural hazards on the transport infrastructure and transport facilities in Russian regions. Using the information collected by the author in the database of technological and natural-technological accidents, contributions of natural factors to road, railway, air, and water transport accident occurrences and traffic disruptions are assessed. All types of natural hazards are considered excluding impacts of global processes (left side in Fig. 1) that are not listed in the database. The risk of road and railway accidents and traffic disruptions, as well as the total risk of transport accidents and disruptions caused by adverse and hazardous natural events is estimated for the area of Russia.

## 2 Materials and methods

### 2.1 Study region


The Russian Federation is the study region.

Federal regions (constituent entities) of the RF were taken as basic territorial units for which all the calculations were performed during the analysis. Federal regions are the main administrative units of the Russian Federation; at this territorial level, all official statistics are published by the Federal State Statistics Service (FSSS) and other

federal institutions of Russia.

The main administrative units of the RF include 85 federal regions: 22 Republics, 9 Territories (Kraies), 46 Regions (Oblast's), 1 Autonomous Region/Autonomous Oblast' (Jewish AR), and 4 Autonomous Areas/Autonomous Okrugs (AO). Moscow, Saint Petersburg, and Sevastopol have a special status of Federal Cities (FC) (cities of federal importance/significance). All the federal regions mentioned in the paper are indicated in Fig. 2.

The size and geographical location of the Russian Federation in various climate and geological conditions determine a great variety of dangerous natural processes and phenomena in its area, including endogenous, exogenous and hydro-meteorological hazards. The most characteristic features of the geography of natural hazards in Russia are as follow:

- Natural hazards associated with cold and snow winters are common throughout the country;

- The population and the economy are relatively low exposed to the most destructive types of natural hazards (earthquakes, tsunamis, hurricanes, etc.), and therefore the frequency of occurrence of natural emergencies with severe consequences is low;

- The historically formed strip of the main settlements from the European part of Russia through the south of Siberia to the Far East approximately coincides with the zone of the smallest manifestation of natural

hazards (Miagkov, 1995).

In Russia, there are several hundred volcanoes, 78 of which are active. Kamchatka and the Kuril Islands are most at risk of volcanic eruptions; explosive eruptions of two to eight volcanoes are observed annually (Girina et al., 2019). About 20 % of the country area with a population of 20 million people is exposed to earthquakes. The most seismically active regions are Kamchatka, Sakhalin, as well as the south of Siberia and the North Caucasus.

Almost the entire territory of Russia is exposed to dangerous exogenous processes; their intensity increases from north to south and from west to east (EMERCOM, 2010). Among exogenous processes, landslides, which are active in 40 % of the country area, debris flows (in 20 %), snow avalanches (in more than 18 % of the area), and other slope processes have the greatest intensity and negative impact on the transport infrastructure. The highest avalanche and debris flow activity is observed in the North Caucasus (Republics of Dagestan, North Ossetia – Alania, and

Kabardino-Balkaria) and in Sakhalin. The greatest intensity of landslides is in the North Caucasus (Chechen, Kabardino-Balkarian, and Karachaevo-Circassian Republics, Republics of Dagestan, Ingushetia, and North Ossetia – Alania; Stavropol and Krasnodar Territories; Rostov Region), Ural (Chelyabinsk and Sverdlovsk Regions), as well as Khabarovsk and Primorye Territories; Amur, Irkutsk, and Sakhalin Regions.

Hydro-meteorological hazardous processes and phenomena such as strong winds, squalls, catastrophic showers,

floods, snowstorms, thunderstorms, hailstorms, etc. are widespread in the country. The combination of heavy precipitation and strong wind is one of the most dangerous climate situations in the coastal regions of the Far East (Sakhalin Region; Kamchatka, Khabarovsk, and Primorye Territories). The highest frequency of strong winds is observed in the south and in the middle part of the European Russia, as well as in the Far East. The most intense rains take place in Kamchatka, Krasnodar, and Primorye Territories; the heaviest snowfalls happen in regions of the

North Caucasus, north and south-west of Siberia, as well as Far East (Sakhalin and Magadan Regions; Chukotka; Kamchatka, Khabarovsk, and Primorye Territories). Regions of the Far East, such as Republic of Sakha (Yakutia), Khabarovsk and Primorye Territories, Amur Region, as well as south of the European Russia (Republics in the North Caucasus; Krasnodar and Stavropol Territories) are mostly exposed to catastrophic floods.

For Russia as a whole, the cumulative degree of natural hazard is increasing from west to east and south, with progress to the mountainous regions. The most dangerous areas in terms of manifestations of natural hazards are situated in the North Caucasus, Ural and Altai Mountains, Irkutsk Region and Trans-Baikal Territory, the Pacific coast of the Far East (Khabarovsk Territory and Magadan Region), and especially Sakhalin, the Kuril Islands, and Kamchatka (Malkhazova and Chalov, 2004).

According to the assessment by EMERCOM (2010), the following federal regions: Republics of Sakha (Yakutia), Karelia, and Komi; Khabarovsk and Primorye Territories; Amur, Arkhangelsk, Irkutsk, Magadan, Murmansk, and Volgograd Regions, as well as Yevish AR, Khanty-Mansi Autonomous Area - Yugra, and Chukotka AO are the most vulnerable to the impacts of natural hazards. The vulnerability is measured as ratio of the total number of realized natural sources of emergencies to the number of emergency situations caused by them. In the listed regions, the vulnerability is higher than an average for Russia.

**2.2 Methodology**

An assessment was made of the risk of road and railway accidents and traffic disruptions, as well as the total risk of transport accidents and disruptions caused by adverse and hazardous natural impacts on the transport infrastructure in Russian federal regions. Road, rail, air, and water transport were considered in the total risk analysis.

Risk is understood as the possibility of undesirable consequences of any action or course of events (Miagkov, 1995).

Risk is measured by the probability of such consequences or the probable magnitude of losses.

There are various methods for assessing risk. In the field of natural hazards, risk is generally defined as by the product of hazard and vulnerability, i.e. a combination of the damageable phenomenon and its consequences (Eckert et al., 2012). The most researchers calculate risk (R) as a function of hazard (H), exposure (E) and vulnerability (V): R=f(H, E, V) (e.g. Arrighi et al., 2013; Falter et al., 2015; IPCC, 2012; Schneiderbauer and Ehrlich, 2004). Various

authors propose their own techniques of calculating risk, mainly within the framework of this common approach. In a recent publication, Arosio et al. (2020) propose a holistic approach to analyze risk in complex systems based on the construction and study of a graph modeling connections between elements.

Another one approach to measuring risk suggests using the concept of emergency situation. In Russia, an emergency situation is defined as a disturbance of the current activity of a populated region due to abrupt technological/natural

impacts (catastrophes or accidents) resulting in social, economic and/or ecological damage, which requires special management efforts to eliminate it (Petrova, 2005). An emergency situation caused by the impact of natural hazards on technological systems and infrastructure can be considered as a result of all the factors of risk: hazard, exposure and vulnerability. It combines hazard defined in its physical parameters, exposure of a population or facilities located in a hazard area and subject to potential losses, and vulnerability that links the intensity of a hazard to

undesirable consequences. An emergency resulting from a hazardous impact may be a measure of the losses due to this impact. The total frequency of emergencies of varying severity may serve as a comprehensive indicator of risk assessment (Shnyparkov, 2004).

In this study, the above approach using frequency of emergency situations as a measure of risk was applied. As an indicator of risk, the average frequency of occurrence of transport accidents and traffic disruptions triggered by

natural hazard impacts, which led to emergency situations of different scale and severity, was used. Risk indicators

were calculated for each federal region as average annual numbers of emergency situations in each type of transport, as well as a resulting average annual number of emergencies due to all transport accidents and disruptions. Thus, the calculated indicators included the probability of undesirable consequences (emergencies) due to impacts of natural hazards on transport infrastructure exposed and vulnerable to these influences. Quantitative and qualitative criteria for classifying transport accidents and disruptions as emergency situations are listed below.

The information collected by the author in an electronic database of technological and natural-technological accidents (created using Microsoft Access) is analyzed in this study. Figure 3 shows the relational structure of the database and the procedure for conducting data analysis.

The database is constantly updated with new information (Petrova, 2011). Currently, it contains about 20 thousand events from 1992 to 2018. Official daily emergency reports of the EMERCOM[1] of Russia and media reports serve as data sources. Only open data is used. Emergency reports are publicly available on the EMERCOM website (https://www.mchs.gov.ru, last access: 20 June 2020), but only in Russian.

The format of the database makes it possible to structure the collected information and classify it according to the author's assessment. The main database table, into which all the information is entered, has the following structure (the listed sections correspond to the column names of the table in Fig. 3):

1) event number: the number changes automatically as information is entered;
2) date of the incident;
3) country;
4) region;
5) location: the distance to the nearest settlement is additionally indicated;
6) type of accident: according to the EMERCOM classification and assessment by the author;
7) a brief description of the event, including the time of occurrence, probable cause of the accident if available, its consequences and measures taken to eliminate them;
8) geographical coordinates if applicable;
9) the scale of the emergency situation caused by the accident: local, inter-municipal, regional, inter-regional, cross-border;
10) the number of deaths;
11) the number of injuries;
12) economic and environmental losses if any;
13) source of information.

All types of technological accidents occurring in Russia are recorded in the database, including those triggered by impacts of natural events of various genesis. Such accidents in technological systems and infrastructure due to natural impacts are classified as natural-technological. The transport accidents and traffic interruptions caused by natural hazards are also listed.

It should be noted that it is not possible to fully cover all the accidents in the database, because they are too numerous, especially road accidents. According to the State traffic inspectorate of the Ministry of Internal Affairs of Russia, 168 thousand road accidents are registered in the RF in 2019.

The criteria for statistical accounting and reporting information about transport accidents by the EMERCOM of Russia are as follows:

1) for road accidents:

---

[1] The Ministry of the Russian Federation for Civil Defense, Emergencies and Elimination of Consequences of Natural Disasters

- any fact of an accident during the transportation of dangerous goods;
- damage to ≥10 motor units;
- traffic interruptions for 12 h due to an accident;
- severe accidents with the death of ≥5 people or injured ≥10 people;

2) for railway accidents:
- any fact of the train crash;
- damage to wagons carrying dangerous goods, causing people to be injured;
- traffic interruptions: on the main railway tracks – for 6 h or more; in the subway – for 30 min and more;

3) for air transport accidents – any fact of the aircraft fall or destruction;

        4) for water transport accidents:
- emergency release of oil and oil products into water bodies in the amount of ≥1 t;
- accidental ingress of liquid and loose toxic substances into water bodies exceeding the maximum permissible concentration by ≥5 times;

- any fact of flooding or throwing of ships ashore as a result of a storm (hurricane, tsunami), landing of ships aground;
- accidents on small vessels with the death of ≥5 people or injured ≥10 people;
- accidents on small vessels carrying dangerous goods.

The same selection criteria are used for events to be included into the author's database. Events that meet these

criteria are characterized as emergency situations.

The accumulation of all the information in the form of an electronic database allows conducting various thematic search queries and analyzing their results depending on the goals and objectives of the research (Fig. 3).

For the purposes of this study, a search of information about transport accidents and traffic disruptions caused by the impacts of natural hazards was made. Road, rail, air, and water transport were included in separate search queries.

Statistical and geographical analysis of data obtained as a result of these search queries was carried out.

The proportion of accidents and disruptions triggered by natural factors was evaluated. All types of natural hazards and adverse weather conditions were taken into account. The main natural causes of accidents and failures were identified for each mode of transport.

Additionally, all the federal regions were divided into groups according to their risk level. The risk level was

estimated for each federal region and each type of transport by the average annual number of emergency situations in comparison with the average value of the indicator in Russia. The number of groups was determined in each case depending on the dispersion of the calculated value. For the analysis, the period from 1992 to 2018 was chosen, since it covered data accumulated in the database.

Using the cartogram method, maps were created, on which the results of the assessment were presented (Fig. 4-6).

**3 Results**

**3.1 Contributions of natural hazards**

The transport infrastructure of Russia is exposed to multiple impacts of various natural hazards and weather phenomena such as heavy rains and snowfalls, strong winds, floods, earthquakes, volcanic eruptions, landslides, debris flows, snow avalanches, rock falls, icing conditions of roads, and others. In many cases, these impacts occur

simultaneously or successively, one after another, and reinforce each other. Some natural hazards trigger hazards of other types, e.g. earthquake or volcanic eruption can provoke such slope processes as rock falls, ice collapses, landslides, debris flows/lahars, snow avalanches, and others; heavy rain can cause debris flows, landslides or floods, etc. Gill and Malamud (2016) examine hazard interrelationships in more detail. These triggering impacts are also recorded in the database and taken into account in the analysis.

Contributions of various natural factors to occurrences of different types of transport accidents and traffic disruptions including road, railway, air, and water transport were found as results of relevant searches in the database. Table 1 shows these results. The "+" sign marks impacts of natural hazards listed in the first column on the corresponding type of transport. Only accidents and disruptions occurred in Russia and recorded in the database are taken into consideration.

As the analysis of the database revealed, transport infrastructure of Russia is most often affected by adverse impacts of meteorological and hydrological origin, especially by hazards associated with cold and snow winters, as well as exogenous slope processes including those provoked by the hydro-meteorological hazards. The majority of emergency situations due to natural hazards are registered from November to March (>67 %); among the warmer months, the largest number of transport accidents occurs in July.

The frequencies of occurrence of accidents and disruptions caused by the impacts of natural hazards, as well as their proportion among other factors of accidents are discussed in the following sections.

### 3.1.1 Road transport

Road transport is one of the main means of moving passengers and goods over short and medium distances in Russia. In terms of transport security, it is the most dangerous means of transportation with the highest number of

fatalities and injuries in accidents (Petrova, 2013) and one of the most common sources of technological hazard, as the number of cars on roads increases significantly faster than the quality of road infrastructure (EMERCOM, 2010). More than 20 % of road accidents and traffic disruptions registered in the database were caused by the impacts of various natural hazards. This refers to those incidents where natural impact was indicated as the main cause of the accident.

Road transport facilities and road infrastructure are exposed to adverse and hazardous natural processes and phenomena of hydro-meteorological character practically all around Russia. Many sections of roads, bridges and other road infrastructure are subject to impacts of snowfalls and snowstorms, heavy rainfalls, flooding, and icing roads; from among exogenous hazards, landslides, debris flows, snow avalanches, rock falls, and other natural hazards affect road infrastructure. These negative impacts trigger road accidents and traffic disruptions leading to

emergency situations and causing many social problems. Under unfavorable meteorological conditions, the risks of car crashes as well as the delay of transportation are increasing, whereas the speed of traffic flow is decreasing (Petrova and Shiryaeva 2019).

During the study period from 1992 to 2018, the following natural hazard impacts that caused accidents and traffic disruptions are identified. They are recorded in 70 from 85 federal regions of Russia. The brackets indicate the

regions where these accidents and failures occurred:

- ***heavy snowfall and snowdrift*** (Republic of Altai; Altai, Kamchatka, Khabarovsk, Krasnodar, Krasnoyarsk, Primorye, and Stavropol Territories; Jewish AR; Yamal-Nenets AO; Amur, Arkhangelsk, Astrakhan, Chelyabinsk, Magadan, Murmansk, Novosibirsk, Omsk, Orenburg, Rostov, Sakhalin, Saratov, Sverdlovsk, and Volgograd Regions);

• ***bottom snowstorm*** (Bashkortostan and Komi Republic; Altai, Kamchatka, and Krasnoyarsk Territories; Chelyabinsk, Magadan, Murmansk, Orenburg, Sakhalin, Ulyanovsk, and Volgograd Regions);

     • ***ice phenomena*** (Republics of Bashkortostan, Kalmykia, and Khakassia; Primorye and Khabarovsk Territories; Jewish AR; Chelyabinsk, Leningrad, Magadan, Rostov, and Sakhalin Regions);

• ***abnormally low air temperature*** (Yamal-Nenets AO; Krasnoyarsk Territory; Kemerovo, Novosibirsk, Omsk, and Tomsk Regions);

     • ***flooding of road due to heavy rain*** (Moscow FC; Republics of Altai, Bashkortostan, Buryatia, Khakassia, Sakha (Yakutia), and Tuva; Chukotka AO; Altai, Krasnodar, Primorye, and Stavropol Territories; Amur, Arkhangelsk, Leningrad, Magadan, Moscow, Nizhny Novgorod, Novgorod, Sakhalin, and Saratov Regions);

• ***washout of road*** (Republic of Sakha (Yakutia); Kamchatka Territory; Sverdlovsk and Tyumen Regions);

     • ***debris flow*** (Chechen, Kabardino-Balkarian, Karachaevo-Circassian, and North Ossetia - Alania Republics; Krasnodar Territory; Sakhalin Region);

     • ***snow avalanche*** (Republics of Dagestan and North Ossetia - Alania);

     • ***rock fall*** (Republics of Dagestan and North Ossetia - Alania);

• ***volcanic eruption*** (Kamchatka Territory).

The majority of all the emergencies revealed (almost 73 %) happened during the cold season from November to March. A significant increasing in their number occurred during abrupt changes in weather conditions, such as heavy precipitation, temperature drops, icing. Emergency situations caused by snow related natural hazards were most often and most common. Snow drifts on the roads became a real disaster leading to long-term traffic

disruptions in many regions of Russia, especially in Arkhangelsk, Chelyabinsk, Novosibirsk, Omsk, Orenburg, Rostov, Sakhalin, and Sverdlovsk Regions; Altai, Khabarovsk, and Krasnodar Territories.

The frequencies of occurrence of road accidents and disruptions due to natural hazards are discussed in Sect. 3.2.1.

### 3.1.2 Railway transport

In the Russian Federation, due to its vast and extended territory and natural features, a large distance of the raw

material base from processing enterprises, railway transportation is the basis of the transport system. It accounts for >80 % of the freight turnover of all types of transport (without pipelines) and >40 % of the passenger traffic of public transport in long-distance and suburban communications. Railway transport is considered the safest form of modern transportation, although railway catastrophes with a large number of victims and injuries occur in many countries. The main causes of railway accidents in Russia are technical problems, a high degree of depreciation (of

tracks, rolling stocks, signaling means, and other equipment) and a "human factor" such as errors of dispatchers and drivers, etc. (Petrova, 2015).

More than 7 % of all railway accidents and failures registered in the database were triggered by natural factors. This refers to those incidents where natural impacts were indicated as the main causes of accidents. Over 1992 to 2018, impacts of natural hazards of various genesis caused railway accidents and traffic disruptions in 29 from 85 federal

regions of Russia.

The identified natural hazards that caused these harmful events are listed below. The brackets indicate the regions where these accidents and failures occurred:

     • ***heavy snow*** (Yamal-Nenets AO; Orenburg and Sakhalin Regions);

- ***washout of railway as a result of heavy rain and flash flood*** (Republics of Dagestan and Karelia, Chuvash and Udmurtian Republics; Khabarovsk and Krasnodar Territories; Amur and Sakhalin Regions);
- ***snow avalanche*** (Khabarovsk Territory; Sakhalin Region);
- ***rails deformation due to heat wave*** (Republic of Kalmykia; Rostov Region);
- ***landslide*** (Krasnodar Territory; Orel Region);
- ***debris flow*** (Krasnodar Territory; Sakhalin Region);
- ***rock fall*** (Republic of Bashkortostan; Khabarovsk and Krasnodar Territories);
- ***flooding due to melting snow*** (Murmansk and Vologda Regions).

Regarding seasonality of accidents, they had two peaks: in summer (in June and July) and in November. The most part of emergency situations were caused by snow drifts, washout or flooding of railway tracks due to heavy rains or floods, as well as by the slope processes such as landslides, snow avalanches, debris flows, and rock falls.

The frequencies of occurrence of railway accidents due to natural hazards are discussed in Sect. 3.2.2.

### 3.1.3 Air transport

Air transport is the fastest and most expensive mode of transportation. That is why it is primarily used to transport passengers over distances of more than 1,000 km. In many distant areas of Russia (in the mountains, in the Far North), it is the only means of transport. The main causes of accidents are technical failures or "human errors", as well as various natural factors including adverse weather or collision with a flock of birds (EMERCOM, 2010).

The adverse weather conditions and other natural hazard impacts caused more than 8 % of all the air transport accidents and traffic disruptions recorded in the database. This refers to those incidents where natural impacts were indicated as the main causes of accidents. Over 1992 to 2018, these events were registered in 27 from 85 federal regions of Russia.

The following impacts of natural hazards were revealed:
- ***strong winds*** (Moscow FC; Republics of Bashkortostan and Tatarstan, Chuvash Republic; Kamchatka, Krasnodar, and Krasnoyarsk Territories; Irkutsk, Murmansk, Omsk, Rostov, Sakhalin, Saratov, and Ulyanovsk Regions);
- ***thunderstorms*** (Republic of Sakha (Yakutia); Irkutsk Region);
- ***heavy rains*** (Moscow FC; Khabarovsk and Krasnodar Territories; Irkutsk Region);
- ***snowfalls and snowstorms*** (Moscow FC; Republic of Khakassia; Kamchatka, Krasnodar, and Krasnoyarsk Territories; Leningrad, Magadan, Rostov, and Sakhalin Regions);
- ***sleets*** (Moscow and St. Petersburg FC; Republics of Bashkortostan and Tatarstan, Chuvash Republic; Kamchatka and Krasnodar Territories; Rostov Region);
- ***runway icing*** (Moscow FC; Kamchatka and Primorye Territories; Kaluga and Murmansk Regions);
- ***fog*** (Moscow FC; Chechen and Ingushetia Republics; Sverdlovsk Region);
- ***snow avalanche*** (Kamchatka);
- ***volcanic eruption***.

In many cases, these adverse impacts occurred simultaneously. Thus, the majority of emergency situations were caused by the combination of heavy snow and strong winds. Almost 66 % of events occurred during the cold season from November to March; another one peak of accidents was in July.

A unique incident, when a helicopter was damaged as a result of an avalanche, was recorded in the database on 10 April 2010 in Kamchatka.

For the study period, there was not a single accident caused by volcanic eruption in Russia. Due to the eruption of the Icelandic volcano Eyyafyatlayokudl, airlines canceled and delayed more than 500 flights at 10 Russian airports in April 2010; 32 thousand passengers could not fly.

The frequencies of occurrence of air transport accidents caused by natural hazards are discussed in Sect. 3.2.3 and included in the total risk analysis (Sect. 3.2.5).

### 3.1.4 Water transport

Water transport includes both sea and river transport. Despite the relatively low speed and seasonal limitations on traffic, this type of transport is widely used for transporting large volumes of goods and passengers at different distances. The main causes of accidents in water transport are violations of the rules of navigation and transportation, of fire safety and technical operation of vessels; depreciation of ships, ports' equipment, and other objects of infrastructure, as well as impacts of natural hazards and adverse weather conditions (EMERCOM, 2010).

The greatest contribution of natural factors to the accident rate after road transport was recorded for water transport. Almost 16 % of all the water transport accidents registered in the database were caused by various natural hazards. These events were registered in 21 from 85 federal regions of Russia.

The following impacts were revealed from 1992 to 2018:

- *strong winds* (Kamchatka, Krasnodar, and Primorye Territories; Leningrad, Sakhalin, and Sverdlovsk Regions);
- *storms* (Republics of Dagestan, Karelia, and Tatarstan; Yamal-Nenets AO; Kamchatka, Khabarovsk, Krasnodar, and Primorye Territories; Astrakhan, Irkutsk, Magadan, Murmansk, Rostov, Ryasan, Sakhalin, and Yaroslavl Regions);
- *snowstorms* (Irkutsk and Sakhalin Regions);
- *icing* (Republic of Sakha (Yakutia); Primorye Territory; Sakhalin Region);
- *thunderstorms* (Komi Republic; Leningrad Region);
- *fog and mist* (Leningrad and Sakhalin Regions).

The most part of accidents (>70 %) occurred during the cold season from September to January.

The frequencies of occurrence of water transport accidents due to natural hazards are discussed in Sect. 3.2.4 and included in the total risk analysis (Sect. 3.2.5).

### 3.2 Risk of transport accidents and traffic disruptions

Occurrence frequencies of road, railway, air, and water accidents and traffic disruptions due to natural hazard impacts at the level of Russian federal regions were estimated for the risk analysis. As mentioned in Sect. 2.2, only accidents and disruptions, which reached the scale of emergency situation, were taken into account. Annual average numbers of such events over 1992 to 2018 were used as risk indicators.

All the federal regions were divided into groups by their risk levels of road and railway accidents, as well as the total risk of transport accidents and traffic disruptions. In each case, the risk level was determined in comparison with the average value of the corresponding indicator for Russia.

The resulting maps were created and analyzed. Regional differences in the risk of transport accidents were found. Below are the main results of the risk analysis.

### 3.2.1 Road transport

Risk of emergencies in road transport depends on the density of the road network, traffic intensity, human factors (violation of traffic rules by drivers and pedestrians, etc.), as well as climatic conditions, seasonality, and other circumstances. With a large area of the country, the paved public road density in Russia is the lowest of all the G8 countries, equal to 63 km per 1,000 km$^2$ (FSSS, 2020). However, it is much higher in the densely populated regions of the European part of Russia. In the Asian part, only some south-western and south-eastern regions have a satisfactory network of hard-surface roads (Petrova and Shiryaeva, 2019). Moscow and St. Petersburg have the highest density of paved public roads, which comprises to about 2,500 km/1,000 km$^2$; it is also high in federal regions of the central Russia (Moscow and Belgorod Regions) and the North Caucasus (Republics of Ingushetia and North Ossetia - Alania), equal to 700–850 km/1,000 km$^2$ (FSSS, 2020).

Risk of road accidents and traffic disruptions due to natural hazard impacts within the Russian federal regions was assessed.

For the risk analysis, 635 emergency situations of various scale and severity caused by the impacts of natural hazards on road infrastructure were taken into consideration. The main triggers of these emergencies and the regions of their occurrence were identified in Sect. 3.1.1. The risk indicator was calculated as an average annual number of emergency situations of this type in each federal region as well as the average for Russia.

All the federal regions are divided into five groups in accordance with risk level by comparing their risk indicators with the average for Russia. Figure 4 shows the resulting map.

Regions of the Far East of Russia (Kamchatka and Khabarovsk Territories, Magadan and Sakhalin Regions), Krasnoyarsk Territory in the southern part of Central Siberia, and Republic of North Ossetia - Alania in the North Caucasus have the highest risk level. The road infrastructure in these regions is mostly affected by the above listed natural hazards, especially by heavy snowfalls and snowstorms, ice phenomena, abnormally low air temperature, and heavy rains. In North Ossetia – Alania, impacts of snow avalanches and debris flows are the most significant.

### 3.2.2 Railway transport

Risk of emergencies in railway transport depends on the density of the railway network, traffic intensity, human factors, climatic conditions, and seasonality. The highest density of the public railway network is in Federal Cities Moscow (1,921 km/10,000 km$^2$) and St. Petersburg (3,082 km/10,000 km$^2$), as well as federal regions of the central and north-western parts of the European Russia such as Moscow, Kaliningrad, Tula, Kursk, Vladimir, and Leningrad Regions (300-500 km/10,000 km$^2$). With a lack of railways in a large part of the country area, especially in its Asian part, the average density of railways in Russia is 51 km/10,000 km$^2$; in the central part of the European Russia it is 263 km/10,000 km$^2$ (FSSS, 2020).

Risk of railway accidents and traffic disruptions due to natural hazard impacts at the level of Russian federal regions was assessed.

For the risk analysis, 63 emergency situations of various scale and severity caused by the impacts of natural hazards on railway infrastructure were taken into consideration. The main triggers of these emergencies and the regions of their occurrence were identified in Sect. 3.1.2. Occurrence frequencies (annual average numbers) of these events were calculated for each federal region as well as the average for Russia.

All the federal regions are divided into three groups by their risk levels. In this case, only three groups are chosen, since the number of accidents and dispersion of risk indicators are not as great as in the case of road accidents. Figure 5 shows the resulting map.

Krasnodar Territory in the southern part of European Russia and regions of the Far East (Sakhalin Region and Khabarovsk Territory) are characterized by the highest level of risk. Railways in these regions are mostly affected by the impacts of heavy snowfalls, heavy rains, snow avalanches, landslides, debris flows, and rock falls.

### 3.2.3 Air transport

Risk of emergencies in air transport depends on the aircraft technical condition, air traffic intensity, human factors, meteorological conditions, and seasonality.

The number of air transport accidents and traffic disruptions due to impacts of natural hazards was included in the calculation of the total risk indicator. For the risk analysis, 70 emergency situations were taken into consideration. The main triggers of these emergencies and the regions of their occurrence were identified in Sect. 3.1.3.

### 3.2.4 Water transport

Risk of emergencies in water transport depends on technical conditions of vessels, traffic intensity, human factors, climatic conditions, and seasonality.

Water transport accidents due to natural impacts were also included in the calculation of the total risk of transport accidents and disruptions. For the risk analysis, 70 emergency situations were taken into consideration. The main

triggers of these emergencies and the regions of their occurrence were identified in Sect. 3.1.4.

### 3.2.5 The total risk

Additionally, the total risk of transport accidents and traffic disruptions was assessed for the area of Russia. Occurrence frequencies of accidents and disruptions in all the above examined types of transport over 1992 to 2018 were used as risk indicators.

For the total risk analysis, 838 emergency situations of various scale and severity caused by the impacts of natural hazards on transport infrastructure were taken into consideration. The main triggers of these accidents were identified in Sect. 3.1 and shown in Table 1; annual average numbers of these events were calculated for each federal region as well as the average for Russia.

All the federal regions were divided into five groups by their risk levels. The procedure for selecting groups was

described in Sect. 2.2.

Figure 6 shows the resulting map. Regions of the Far East (Kamchatka, Khabarovsk, and Primorye Territories; Magadan and Sakhalin Regions), Krasnoyarsk Territory in the south part of Central Siberia, Murmansk Region in the north and Krasnodar Territory in the south part of European Russia and Republic of North Ossetia - Alania in the North Caucasus have the highest level of risk. The transport infrastructure in these regions is mostly affected by the

adverse impacts of natural hazards listed in Table 1, primarily those of hydro-meteorological genesis. Kamchatka, Khabarovsk, and Primorye Territories, as well as Sakhalin Region are characterized by the most dangerous meteorological combinations of heavy precipitations and strong winds. In Kamchatka, Krasnodar, and Primorye Territories, the most intense rains are recorded. In winter, the heaviest snowfalls happen in all the above regions. In spring and early autumn, Khabarovsk, Krasnodar, and Primorye Territories are subject to catastrophic floods.

Kamchatka is most at risk of volcanic eruptions. Republic of North Ossetia - Alania and Sakhalin Region are characterized by the highest avalanche and debris flow activity. All of the mentioned natural hazards trigger accidents and lead to delay in the transportation of passengers and goods by road, railway, air, and water transport. In addition, Kamchatka, Sakhalin, south part of Siberia, and the North Caucasus are among the most seismically

active regions of Russia; during the study period, no traffic accidents due to the earthquake were recorded, but their
possibility should be taken into account.

## 4 Concluding remarks and discussion

Contributions of various natural hazards to occurrences of different types of transport accidents and traffic disruptions including road, railway, air, and water transport are revealed. Among all the identified types of natural hazards, hydro-meteorological hazards such as heavy snowfalls and rains, floods and ice phenomena, as well as
dangerous exogenous slope processes including snow avalanches, debris flows, landslides, and rock falls have the largest contributions to transport accidents and disruptions. The most dangerous is the combination of heavy precipitations and strong winds.

An annual average frequency of occurrences of emergency situations of various scale and severity is applied in this study among all possible methods for assessing risk. Unlike methods that assess risk by measuring its components
such as hazard, exposure, and vulnerability, this approach takes into account the resulting consequences of the above factors and the probability of these consequences. Transport accidents and disruptions are considered in this case as consequences of natural hazard impacts on transport infrastructure that is exposed and vulnerable to these impacts. The risk index is calculated as an annual average number of emergency situations caused by natural hazard impacts in each federal region and each type of transport. Thus, the index used combines both the probability and severity of
the adverse impacts of natural hazards on transport infrastructure, as well as vulnerability of infrastructure to these adverse impacts resulting in accidents and malfunctions. Using this method, it is possible to compare between different regions and identify deficiencies that need to be addressed.

Regional differences in the risk of transport accidents between Russian federal regions were found. All the federal regions were divided into groups by their risk levels of road and railway accidents, as well as the total risk of
transport accidents and traffic disruptions due to natural hazard impacts. The resulting maps were created and analyzed.

Kamchatka, Khabarovsk, Krasnodar, Krasnoyarsk, Primorye Territories; Magadan, Murmansk, and Sakhalin Regions, and Republic of North Ossetia - Alania are characterized by the highest risk of transport accidents and traffic disruptions caused by natural events. Emergencies of various scales occur in these regions on average more
often than once a year (Fig. 6). Chelyabinsk, Orenburg, and Rostov Regions, Altai Territory, Republics of Dagestan and Bashkortostan, and Moscow have a high risk level with an average probability of 1 event in 1-2 years (0.6-1.0 events per year).

For the study period of 1992 to 2018, the database mainly recorded events caused by hydro-meteorological and exogenous natural hazards. With high value of the risk index, Kamchatka, Sakhalin, the North Caucasus, and south
of Siberia are also among the most seismically active regions of Russia, which further increases the likelihood of emergencies in these regions in case of an earthquake. It is in these regions that the necessary measures should first be taken to reduce the vulnerability of transport infrastructure to undesirable natural impacts and increase level of protection and preparedness.

Under conditions of observed and forecasted global and regional climate changes, adverse and hazardous natural
impacts on various facilities of transport infrastructure, primarily from natural hazards of meteorological and hydrological origin, as well as other natural events triggered by them such as landslides, snow avalanches, and debris flows are expected to increase (Malkhazova and Chalov, 2004; Yakubovich et al., 2018). Other factors, such as growing transportation network, increased traffic and the lack of funding will also lead to increasing of adverse

impacts, especially with further development of transport infrastructure to areas with high level of natural risk. In this regard, continuous monitoring and assessment of natural hazard impacts is especially relevant and important.

Only severe accidents leading to an emergency situation were considered in this study due to a lack of data on small events. This gap should be filled in a future research because small events can also cause a great damage to the infrastructure and trigger accidents and traffic interruptions (Voumard et al., 2018).

Effects of global processes such as space weather on the transport infrastructure facilities, especially on electronics and automatic machinery were not taken into consideration because these events were not recorded in the database. In the future, these impacts should be also investigated; risk of these events should be considered in the risk assessment.

**Acknowledgements**

The work described in this paper was supported by Lomonosov Moscow State University (grant I.7 AAAA-A16-116032810093-2 "Mapping, modeling and risk assessment of dangerous natural processes").

**Data availability:**

The data used in this study are collected by the author in an electronic database, which is not available publicly.

**Competing interest:**

The author declares that she has no conflict of interest.

**Author's contribution:**

The work presented in this study was conducted by E. Petrova.

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

**Table 1: Transport accidents and traffic disruptions caused by natural hazards in Russia (1992-2018)**

| Type of transport / Natural hazard | Road transport | Railway transport | Air transport | Water transport |
|---|---|---|---|---|
| Strong wind, storm | | | + | + |
| Snowfall, snowstorm, snowdrift, sleet | + | + | + | + |
| Rainfall, hailstone | + | + | + | |
| Hard frost, icing, ice-crusted ground | + | | + | + |
| Thunderstorm, lightning | | | + | + |
| Fog, mist | + | | + | + |
| Flood | + | + | | |
| Heat wave | | + | | |
| Earthquake, volcanic eruption | + | | + | |
| Landslide, slump, debris flow | + | + | | |
| Rock fall | + | + | | |
| Snow avalanche | + | + | + | |

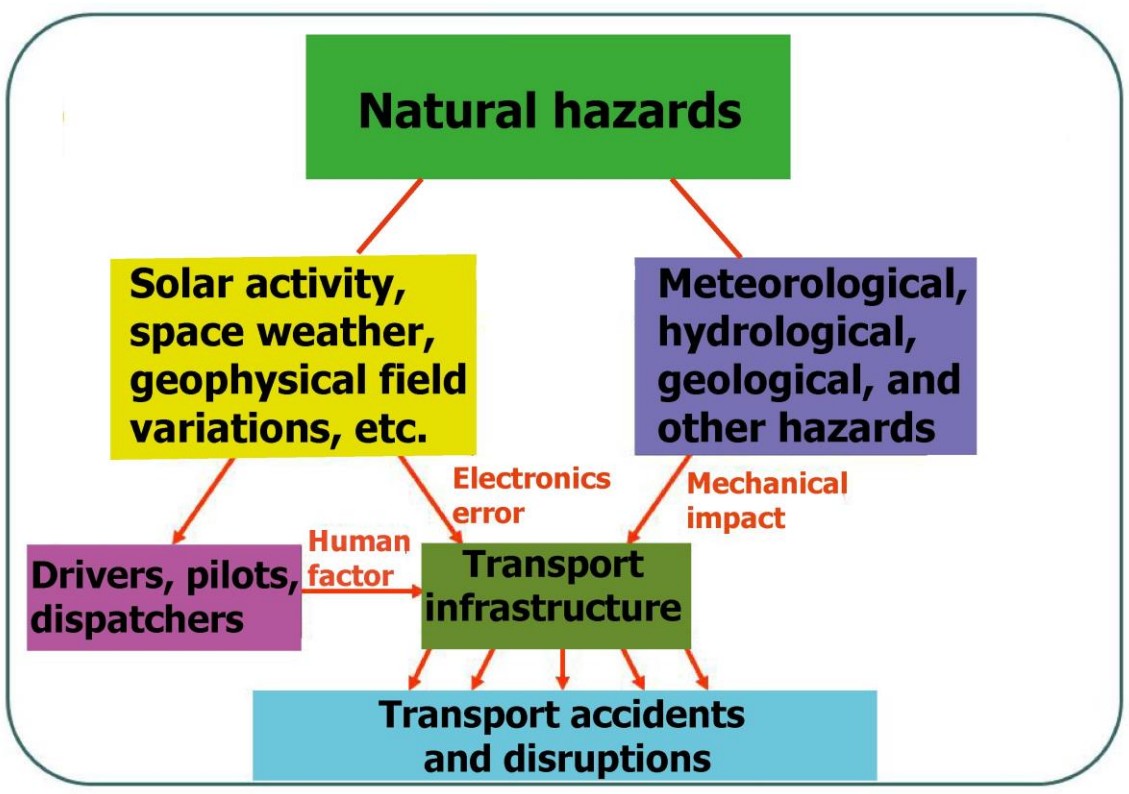


**Figure 1: Grouping of natural hazards based on their genesis and impacts on transport infrastructure**

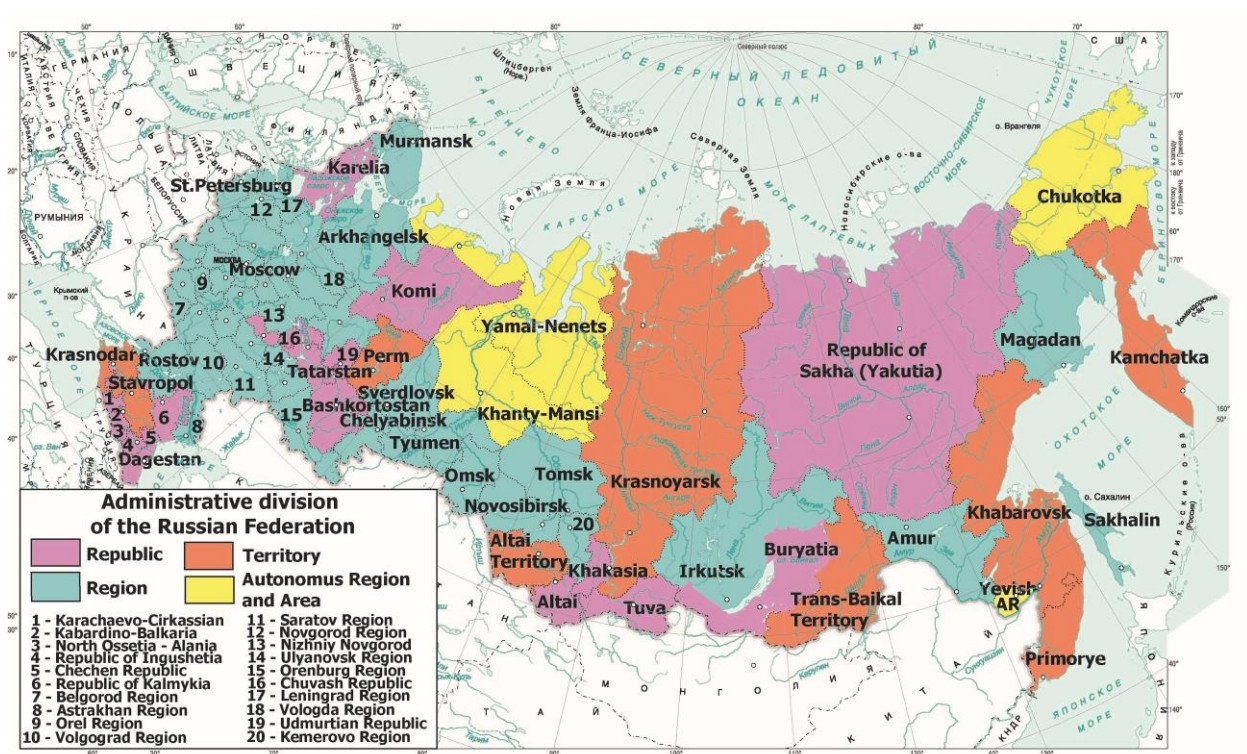

**Figure 2: Federal regions of the Russian Federation**
         **(base map: © DIK - Publishing House: Design. Information. Cartography)**

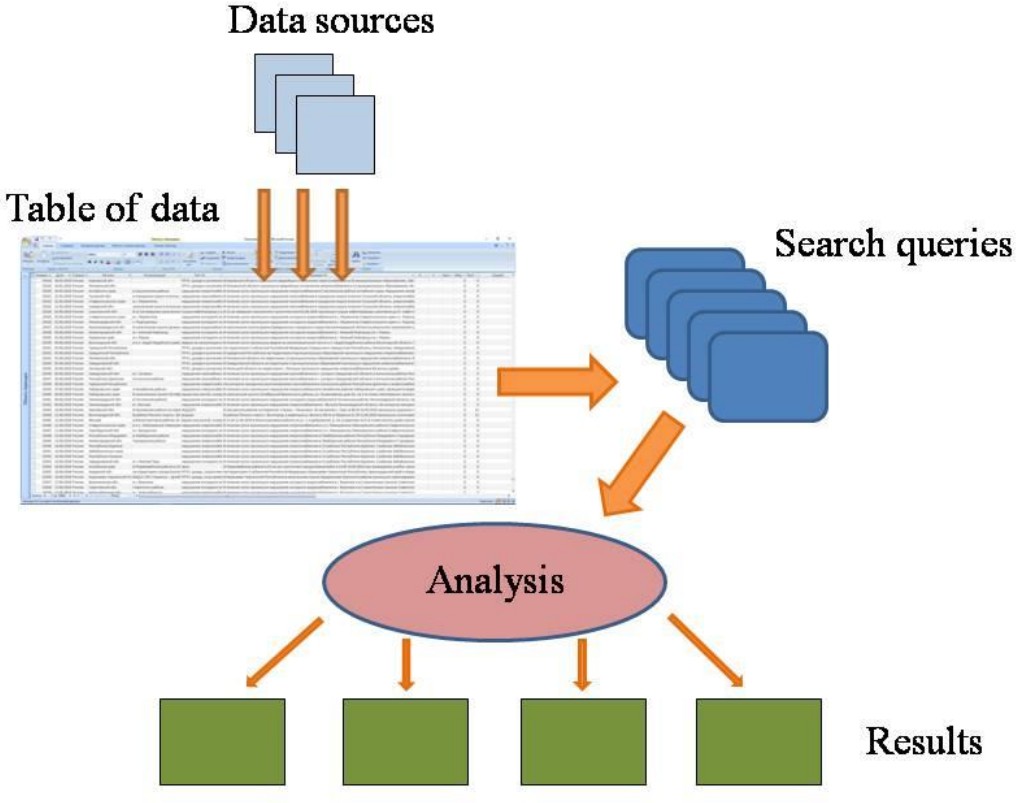

**Figure 3: Relational structure of the database**

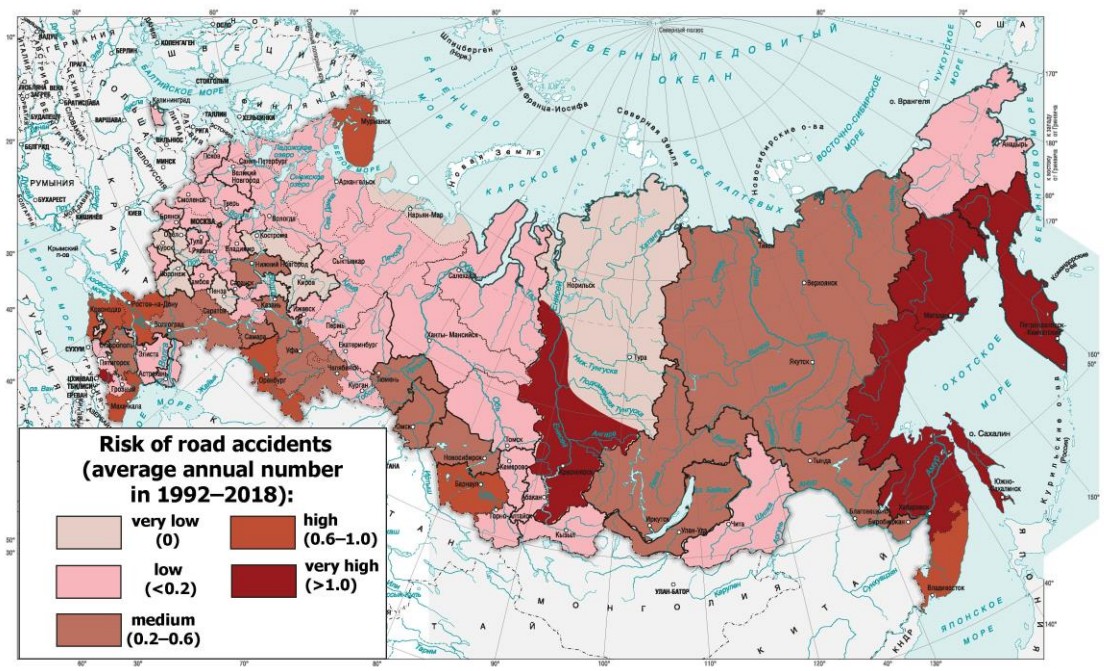

**Figure 4: Risk of road accidents and traffic disruptions triggered by natural hazards in the RF**
**(base map: © DIK - Publishing House: Design. Information. Cartography)**


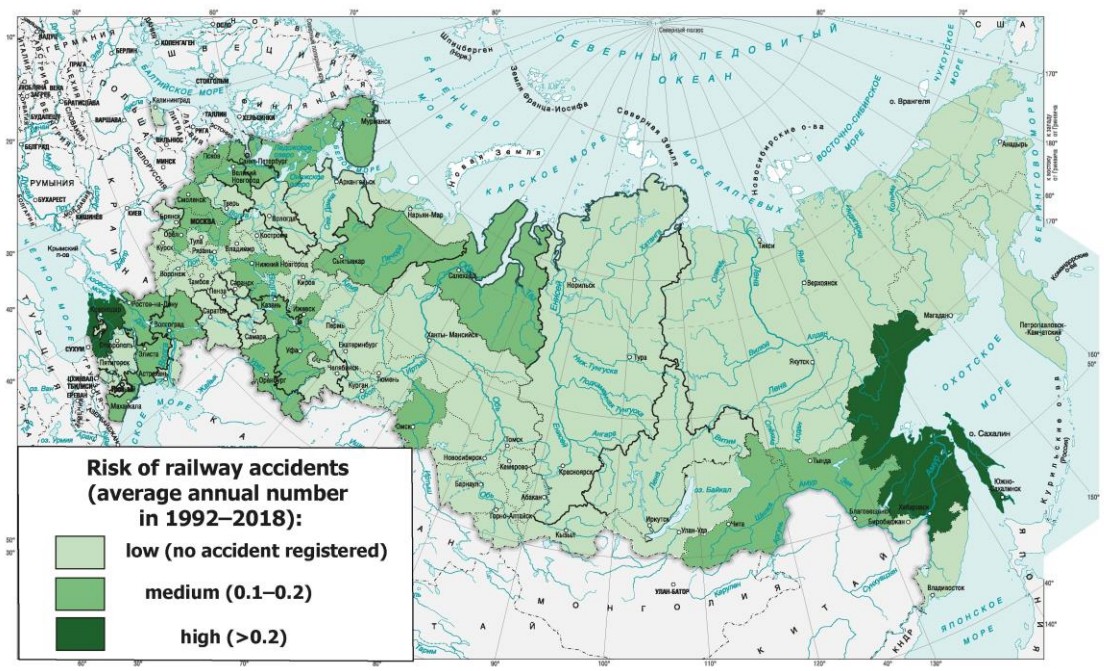

**Figure 5: Risk of railway accidents and traffic disruptions triggered by natural hazards in the RF**
**(base map: © DIK - Publishing House: Design. Information. Cartography)**


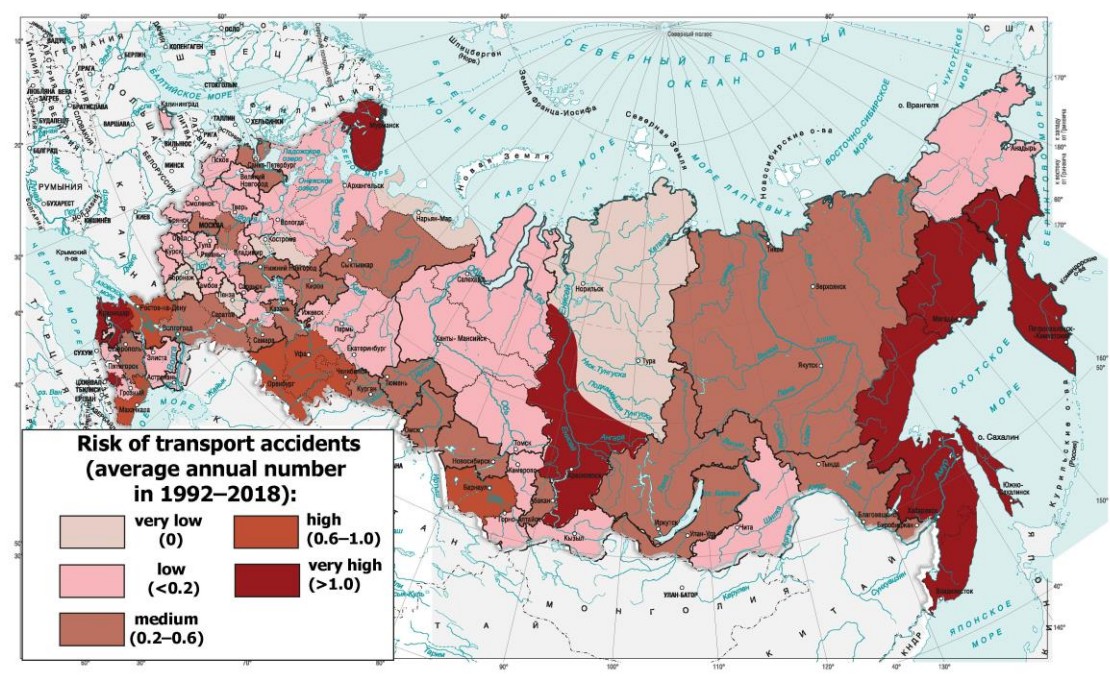

**Figure 6: Risk of transport accidents and disruptions triggered by natural hazards in the RF**
**(base map: © DIK - Publishing House: Design. Information. Cartography)**