# Peer review of "Natural hazard impacts on transport infrastructure in Russia"

_Natural Hazards and Earth System Sciences, 2019_

## Referee Comment (RC1) · Anonymous Referee #1 · 7 Mar 2020

In her paper the author describes an analysis of impacts to the Russian transportation infrastructure due to natural hazards. The analysis is based on a historical database with incidents between 1992 - 2018, which was developed by the author.

Although the general topic of the paper is highly relevant for NHESS there are several major issues which need to be addressed before publication.

The introduction section (section 1) provides an introduction to transportation infrastructure in general and related vulnerabilities due to natural hazards. This section does not have any scientific references related to possible classifications of transportation infrastructure (including subcategories) and natural hazards. For example it remains unclear why the author chose the natural hazard classification presented in figure 1 and not other published classification schemes. The reference in line 33 is missing in the

reference section. The literature review (line 55 ff) is quite comprehensive in the sense that it includes many references, but the analysis with respect to the presented study is very rough and lacks detail. Just a mere listing of references with just a few sentences is not sufficient for a journal paper. But I like that the author looked for papers which described various natural hazard impacts to traffic infrastructures. This needs to be expanded in a revision.

Section 2 is too brief and lacks detail. The study region is only described by region, but no hazard information is provided for those regions. The paper remains on the level of hazard categorization in general. A deeper description of Russia on region level with respect to hazards and vulnerabilities is needed. The methodology section is super brief and it does not sufficient detail about the data sources, the selection criteria / levels for data to be included, the structure of the database, etc. Without this information nobody can reproduce the database or assess the quality of the produced database. There is also no definition of risk and it is unclear how the five risk categories are calculated. Just looking at incidents in a database – even with information about natural hazards – does not qualify for a risk analysis. It is more like a statistical analysis of a database. The author needs to describe the method in a detailed and understandable way and she should also include scientific references in the methodology section.

Section 3 is a qualitative description of natural hazard induced incidents to the transportation sectors road, rail, water and air. As a sub section of an improved paper this may provide valuable insights to better understand the vulnerability of transportation infrastructure in Russia, but without a sound section 2 it remains unclear whether these results make sense or not. Structuring the analysis along the transportations modes is fine and should be kept, but it should be more analytical and not just descriptive.

The conclusion section lacks also detail and it remains unclear what the main contribution of the paper is. A critical reflection on the method is very brief and the discussion could be expanded, but without knowing more about the methodology and the underlying risk analysis the reviewer can not provide any meaningful recommendations for

improvement for this section.

---

## Referee Comment (RC2) · Anonymous Referee #2 · 20 Mar 2020

General comments:

The author presents the impact of natural hazards on various types of transportation networks in the Russian Federation, based on a database containing the important accidents which occurred in the recent years. Besides providing potentially useful statistics (although the database is not publicly available), the author does not make a comprehensive analysis to really evaluate the causes of risks and the correlation between a specific type of hazard, it potential manifestation in time and the direct and indirect vulnerability of the infrastructure, nevertheless providing a risk of transport accidents and disruptions map which in my opinion induces in error. Therefore, I do not recommend the publication of this article in this general form, without major modifications.

Specific comments I attach a pdf with my specific comments, hoping that they will help

to author to redefine the paper.

Please also note the supplement to this comment:
https://www.nat-hazards-earth-syst-sci-discuss.net/nhess-2019-426/nhess-2019-426-RC2-supplement.pdf

**Supplement:**

[revised manuscript text omitted]

---

## Author Comment (AC1) · 25 Apr 2020

I thank Referee#1 for his/her very useful comments. They allowed improving the manuscript. The
reviewer's comments were taken into account in the revised version of the manuscript, as explained
below. The reviewer's comments are in italics, the answers are in black and the changes made to the text
are in red. The lines numbers refer to the lines numbers of the revised manuscript.

**Answers to Reviewer#1 comments**

*In her paper the author describes an analysis of impacts to the Russian transportation infrastructure due*
*to natural hazards. The analysis is based on a historical database with incidents between 1992 - 2018,*
*which was developed by the author. Although the general topic of the paper is highly relevant for NHESS*
*there are several major issues which need to be addressed before publication. The introduction section*
*(section 1) provides an introduction to transportation infrastructure in general and related vulnerabilities*
*due to natural hazards. This section does not have any scientific references related to possible*
*classifications of transportation infrastructure (including subcategories) and natural hazards. For*
*example it remains unclear why the author chose the natural hazard classification presented in figure 1*
*and not other published classification schemes.*

Classification of the transport infrastructure of Russia, which is given in the manuscript, refers to the
Federal Law "On Transport Security". This citation was included in the list of references.

Classification of natural hazards presented in figure 1 was proposed by the author. The explanation was
included in the manuscript. The following paragraphs of the introduction section were modified:

"Natural processes and phenomena can be classified in various ways depending on the objectives of a
study. Natural hazards can be typify according to their genetic features, the intensity of their
manifestation, the main formation and development factors, characteristics of spatial distribution and
mode, etc. (Malkhazova and Chalov, 2004).
Previously, two types of natural hazards were found, based on their genesis, distribution in space and
time, and the impact pattern on the technosphere and society in populated areas (Petrova, 2005). In the
context of the present study, the proposed classification scheme was adapted taking into account impacts
of natural hazards on the transport infrastructure (Figure 1).
Solar and geomagnetic disturbances (space weather), geodynamics, geophysical and astrophysical field
variations, and other global processes belong to the first group. They have global scale in space and cyclic
development in time. Natural processes of this type may influence the transport infrastructure both
directly, causing electronics error and automatic machinery failure, as well as indirectly, by affecting the
nervous system of operators, drivers or pilots and thereby leading to a decrease in their reliability. Natural
hazards of the second type are of more "earthly" origin, i.e. from the atmosphere, lithosphere,
hydrosphere or biosphere. They vary greatly in their spatial scale and geographical location. This type of
natural hazards includes earthquakes, volcanic eruptions, landslides, snow avalanches, hurricanes,
windstorms, heavy rains, hail, lightning, snow and ice storms, temperature extremes, wild fires, floods,
droughts, etc. Natural hazards belonging to this group cause a direct destructive effect leading to
accidents and disruptions." - (Lines 360-380)

*The reference in line 33 is missing in the reference section. –*

The reference in line 33 was presented in the reference section as: Geography, society, environment,
Collective monograph, v. 4: Natural and anthropogenic processes and environmental risk, Moscow,
Gorodets Publishing House, 2004.

This reference was revised; the names of the editors were added: Malkhazova, S. M. and Chalov, R. S.
(Eds.): Geography, Society and Environment. Vol. IV: Natural-Anthropogenic Processes and
Environmental Risk, Gorodets Publishing House, Moscow, Russia, 2004.

*The literature review (line 55 ff) is quite comprehensive in the sense that it includes many references, but*
*the analysis with respect to the presented study is very rough and lacks detail. Just a mere listing of*
*references with just a few sentences is not sufficient for a journal paper. But I like that the author looked*
*for papers which described various natural hazard impacts to traffic infrastructures. This needs to be*
*expanded in a revision.*

In the revised manuscript, the literature review will be modified and expanded as follows:

[revised manuscript text omitted]

*The methodology section is super brief and it does not sufficient detail about the data sources, the selection criteria / levels for data to be included, the structure of the database, etc. Without this information nobody can reproduce the database or assess the quality of the produced database.*

The methodology section was modified; the following paragraphs with more detail about the data sources, the selection criteria for data to be included, and the structure of the database were added to Section 2.2:

"The format of the database makes it possible to structure the collected information and classify it according to the author's assessment. The main database table, into which all the information is entered, has the following structure:

1) event number - the number changes automatically as information is entered;

2)  date of the incident;
3)  country;
4)  region;
5)  location - the distance to the nearest settlement is additionally indicated;
6)  type of accident - according to the EMERCOM classification and assessment by the author;
7)  a brief description of the event, including the time of occurrence, probable cause of the accident,
if available, its consequences, and measures taken to eliminate them;
8)  geographical coordinates, if applicable;
9)  the scale of the emergency situation caused by the accident – local, inter-municipal, regional,
inter-regional, cross-border;
10) the number of deaths;
11) the number of injuries;
12) economic and environmental losses, if any;
13) source of information.

All types of technological accidents occurring in Russia are recorded in the database, including those triggered by impacts of natural events of various genesis. Such accidents in technological systems and infrastructure due to natural impacts are classified as natural-technological. The transport accidents and traffic interruptions caused by natural hazards are also listed." - (Lines 521-542)

"The criteria for statistical accounting and reporting transport accident information by the EMERCOM of Russia are as follows:

1)  for road accidents:
    • Any fact of an accident during the transportation of dangerous goods;
    • Damage to 10 or more motor units;
    • Traffic interruptions for 12 hours due to an accident;
    • Severe accidents with the death of five or more people or injured 10 or more people.
2)  for railway accidents:
    • Any fact of the train crash;
    • Damage to wagons carrying dangerous goods, causing people to be injured;
    • Traffic interruptions: on the main railway tracks – for 6 hours or more; in the subway – for 30 minutes and more;
3)  for air transport accidents – any fact of the aircraft fall or destruction;
4)  for water transport accidents:
    • Emergency release of oil and oil products into water bodies in the amount of 1 ton or more;
    • Accidental ingress of liquid and loose toxic substances into water bodies exceeding the maximum permissible concentration by 5 or more times;
    • Any fact of flooding or throwing of ships ashore as a result of a storm (hurricane, tsunami), landing of ships aground;
    • Accidents on small vessels with the death of five or more people or injured 10 or more people;
    • Accidents on small vessels carrying dangerous goods.
The same selection criteria are used for events to be included into the author's database. Events that meet these criteria are characterized as emergency situations." - (Lines 550-574)

*There is also no definition of risk and it is unclear how the five risk categories are calculated. Just looking at incidents in a database – even with information about natural hazards – does not qualify for a risk analysis. It is more like a statistical analysis of a database. The author needs to describe the method in a detailed and understandable way and she should also include scientific references in the methodology section.*

Definition of risk and a detailed description of the method, as well as scientific references were included in the methodology section:

[revised manuscript text omitted]

*Section 3 is a qualitative description of natural hazard induced incidents to the transportation sectors road, rail, water and air. As a sub section of an improved paper this may provide valuable insights to better understand the vulnerability of transportation infrastructure in Russia, but without a sound section 2 it remains unclear whether these results make sense or not. Structuring the analysis along the transportations modes is fine and should be kept, but it should be more analytical and not just descriptive.*

Section 3 was revised; the changes made to the text are in red in the revised version of the manuscript below (Lines 631-898).

*The conclusion section lacks also detail and it remains unclear what the main contribution of the paper*
*is. A critical reflection on the method is very brief and the discussion could be expanded, but without*
*knowing more about the methodology and the underlying risk analysis the reviewer can not provide any*
*meaningful recommendations for improvement for this section.*

The Conclusion section was revised as follows:

[revised manuscript text omitted]

Legend (inside map):

**Risk of transport accidents (average annual number in 1992-2018):**

- very low (0)
- low (<0,2)
- medium (0,2-0,6)
- high (0,6-1,0)
- very high (>1,0)

**Figure 5**: **Risk of transport accidents and disruptions triggered by natural hazards in the RF**

**(base map: © DIK - Publishing House Design. Information. Cartography)**

---

## Author Comment (AC2) · 25 Apr 2020

I thank Referee#2 for his/her very useful comments. They allowed improving the manuscript. The
reviewer's comments were taken into account in the revised version of the manuscript, as explained
below. The reviewer's comments are in italics, the answers are in black and the changes made to the text
are in red. The lines numbers refer to the lines numbers of the revised manuscript.

*General comments: The author presents the impact of natural hazards on various types of transportation*
*networks in the Russian Federation, based on a database containing the important accidents which*
*occurred in the recent years. Besides providing potentially useful statistics (although the database is not*
*publicly available), the author does not make a comprehensive analysis to really evaluate the causes of*
*risks and the correlation between a specific type of hazard, it potential manifestation in time and the*
*direct and indirect vulnerability of the infrastructure, nevertheless providing a risk of transport accidents*
*and disruptions map which in my opinion induces in error. Therefore, I do not recommend the*
*publication of this article in this general form, without major modifications. Specific comments I attach a*
*pdf with my specific comments, hoping that they will help to author to redefine the paper.*

The manuscript was revised. All changes made to the text are described in detail below.

**Answers to Reviewer#2 specific comments**

*Line 2 - railway – This word is doubled; bus stations are not necessary relevant – the enumeration can be*
*simplified.*

The enumeration was revised; the doubled word was deleted:

"According to the Federal Law "On Transport Security" (2019), transport infrastructure of the Russian
Federation (RF) is considered as a large and complex technological system including
tunnels, overpasses, and bridges;  terminals and stations; river and sea ports;
airports;  roads, railways, and  waterways, as well as other buildings,
structures,  and equipment ensuring the functioning of the transport system." (Lines 350-354)

*Lines 23 – 26 – It's not good to repeat the exact same in the previously mentioned abstract.*

The abstract was revised; sentences that repeated the main text of the manuscript were deleted.

*Line 30 – almost all of the listed facilities - maybe it sounds a bit exagerated?*

I agree with this comment. The paragraph was revised as follows:

" Due to the large length of the
transportation network, as well as climatic, geological, geomorphologic, and other natural features of the
country, transport infrastructure facilities of Russia are exposed to the undesirable impacts of adverse
natural processes and phenomena, as well as natural hazards of various genesis, such as geophysical,
hydro-meteorological, and others . Their distribution through the country area is
discussed below in section 2.1." (Lines 358-363)

*Line 32 – reference not according to journal specifications*

The citation of this reference was revised as follows: (Malkhazova and Chalov, 2004). The names of the
editors were used instead of the title of the book.

*Lines 33 – 34 – Once again, the abstract text is reused – not a good practice in my opinion.*

The abstract was revised; repeating text was deleted.

*Line 55 – The author should be mentioned.*

The author of the Transport Strategy is the Ministry of Transport of the Russian Federation. The citation was modified accordingly.

*Line 67 – If you are talking about the impact of natural hazards, there are numerous statistics (especially in developed countries) providing the causes of accidents – please search for them.*

I agree with the reviewer. The literature review was revised; the changes made are below:

"All the authors agree that the adverse weather is a major factor affecting road situation (e.g. Edwards 1996; Rakha et al 2007; Andrey 2010; Andersson and Chapman 2011; Bergel-Hayat et al 2013; Chakrabarty and Gupta 2013). Many authors connect the maximum number of road accidents with precipitations (Jaroszweski and McNamara 2014; Spasova and Dimitrov 2015). Aron et al (2007) revealed that 14% of all injury accidents in Normandy (France) took place during rainy weather and 1% during fog, frost or snow / hail. Satterthwaite (1976) found the rainy weather to be a major factor affecting accident numbers on the State Highways of California: on very wet days the number of accidents was often double comparing to dry days. Brodsky & Hakkert (1988) with data from Israel and the USA did indicate that the added risk of an injury accident in rainy conditions can be two to three times greater than in dry weather. And when a rain follows a dry spell – the hazard could be even greater. Among other weather factors, bright sunlight was identified as a cause of accidents (Shiryaeva 2016). Redelmeier and Raza (2017) investigated visual illusions created by bright sunlight that lead to driver error, including fallible distance judgment from aerial perspective. According to their results, the risk of a life-threatening crash was 16% higher during bright sunlight than normal weather. Some authors consider other natural hazards, such as landslides (Bíl et al., 2014; Schlögl et al., 2019), flash floods (Shabou et al., 2017) or rock falls (Bunce et al., 1997; Budetta and Nappi, 2013).  As for railway transport, most of papers also focus on specific hazards, considering impacts of adverse weather and hydro-meteorological extremes (Ludvigsen and Klæboe, 2014; Nogal et al., 2016), landsliding (Jaiswal et al., 2011), flooding (Hong et al., 2015; Kellermann et al., 2016), snowfall (Ludvigsen and Klæboe, 2014) or tree falls (Nyberg and Johansson, 2013; Bil et al., 2017) as triggers of accidents. Some studies combine all types of natural hazards affecting road and rail infrastructure (Govorushko 2012; Petrova, 2015; Kaundinya et al., 2016). Voumard et al. (2018) examine small events like earth flow, debris flow, rockfall, flood, snow avalanche, and others, which represent three-quarters of the total direct costs of all natural hazard impacts on Swiss roads and railways.  Investigations of natural hazard impacts on other transport systems than roads and railways are not so numerous. As example, studies about danger of volcanic eruptions to the aviation should be mentioned (Neal et al, 2009; Brenot et al., 2014; Girina et al., 2019). Large explosive eruptions of volcanoes can eject several cubic kilometers of volcanic ash and aerosol into the atmosphere and stratosphere during a few hours or days posing a threat to modern airliners (Gordeev and Girina, 2014)." - (Lines 407-439)

*Line 86 – There are also more recent studies available, such as Donald A. Redelmeier, Shehariar Raza (2017) or Jonathan J.Rolison et al. (2018)*

I thank the reviewer for pointing me to these very interesting studies. The studies by Donald A. Redelmeier, Shehariar Raza (2017) and Jonathan J.Rolison et al. (2018) do not investigate impacts of solar activity on drivers, which are discussed in the manuscript. Donald A. Redelmeier and Shehariar Raza (2017) investigate visual illusions created by bright sunlight that lead to driver error. This is another one aspect. Nevertheless, this reference was included into the literature review. Jonathan J.Rolison et al. (2018) study differences between real factors that contribute to road accidents and factors reported by police officers in accident report forms. They do not take into account impacts of solar activity on drivers among of contributing factors.

*Line 118 – Does large economic damage have a qualitative definition?*

Yes, it has a qualitative definition. The sentence was replaced by the following paragraphs, which include
damage information for each mode of transport: "The criteria for statistical accounting and reporting
transport accident information by the EMERCOM of Russia are as follow:

1) for road accidents:
• Any fact of an accident during the transportation of dangerous goods;
• Damage to 10 or more motor units;
• Traffic interruptions for 12 hours due to an accident;
• Severe accidents with the death of five or more people or injured 10 or more people.
2) for railway accidents:
• Any fact of the train crash;
• Damage to wagons carrying dangerous goods, causing people to be injured;
• Traffic interruptions: on the main railway tracks – for 6 hours or more; in the subway –
for 30 minutes and more;
3) for air transport accidents – any fact of the aircraft fall or destruction;
4) for water transport accidents:
• Emergency release of oil and oil products into water bodies in the amount of 1 ton or
more;
• Accidental ingress of liquid and loose toxic substances into water bodies exceeding the
maximum permissible concentration by 5 or more times;
• Any fact of flooding or throwing of ships ashore as a result of a storm (hurricane,
tsunami), landing of ships aground;
• Accidents on small vessels with the death of five or more people or injured 10 or more
people;
• Accidents on small vessels carrying dangerous goods." - (Lines 556-578)

*Line 120 – In which statistics? Please explain a bit better the difference the data base provides compared*
*to EMERCOM data which I believe is considered also in the statistics.*

The sentence was replaced by the following paragraphs explaining database features:

"The format of the database makes it possible to structure the collected information and classify it
according to the author's assessment." - (Lines 527-528)

"The accumulation of all the information in the form of an electronic database allows conducting various
thematic search queries and analyzing their results depending on the goals and objectives of the research."
- (Lines 581-582)

*Line 146 – Road transport is probably a more comprehensive analysis category.*

I agree with this comment. The word "automobile" was replaced by "road".

*Line 178 – is it correlated the triggering impact of earthquakes on other natural hazards?*

The following explanation was added to section 3.1:

"Some natural hazards trigger hazards of other types, e.g. earthquake or volcanic eruption can provoke
such slope processes as rock falls, ice collapses, landslides, debris flows / lahars, snow avalanches, and
others; heavy rain can cause debris flows, landslides or floods, etc. Gill and Malamud (2016) examine
hazard interrelationships in more detail. These triggering impacts are also recorded in the database and
taken into account in the analysis." - (Lines 643-647)

*Line 226 – Risk should be correlated also with the length of roads in a specific territory, traffic values*
*and moment of day for the occurrence of natural hazards. Without a form of normalisation, it is just*
*statistics and not risk analysis.*

Factors affecting risk of accidents in each type of transport are discussed in the revised version of the
manuscript in sections 3.2.1-3.2.4. The changes made to the text are marked in red in the manuscript.
Definition of risk and a detailed description of the method used were included in the methodology
section:

"Risk is understood as the possibility of undesirable consequences of any action or course of events
(Miagkov, 1995). Risk is measured by the probability of such consequences or the probable magnitude of
losses. There are various methods for assessing risk. In the field of natural hazards, risk is generally
defined as by the product of hazard and vulnerability, i.e. a combination of the damageable phenomenon
and its consequences (Eckert et al., 2012). The most researchers calculate risk (R) as a function of hazard
(H), exposure (E) and vulnerability (V): R=f(H,E,V) (e.g. Arrighi et al., 2013; Falter et al., 2015; IPCC,
2012; Schneiderbauer and Ehrlich, 2004). Various authors propose their own techniques of calculating
risk, mainly within the framework of this common approach. In a recent publication, Arosio et al. (2020)
propose a holistic approach to analyze risk in complex systems based on the construction and study of a
graph modeling connections between elements.
Another one approach to measuring risk suggests using the concept of emergency situation. In Russia, an
emergency situation is defined as a disturbance of the current activity of a populated region due to abrupt
technological / natural impacts (catastrophes or accidents) resulting in social, economic, and / or
ecological damage, which requires special management efforts to eliminate it (Petrova, 2005). An
emergency situation caused by the impact of natural hazards on technological systems and infrastructure
can be considered as a result of all the factors of risk: hazard, exposure and vulnerability; it combines
hazard defined in its physical parameters, exposure of a population or facilities located in a hazard area
and subject to potential losses, and vulnerability that links the intensity of a hazard to undesirable
consequences. An emergency resulting from a hazardous impact may be a measure of the losses due to
this impact. The total frequency of emergencies of varying severity may serve as a comprehensive
indicator of risk assessment (Shnyparkov, 2004).
In this study, the above approach using frequency of emergency situations as a
measure of risk was applied. As an indicator of risk, the average frequency of occurrence of transport
accidents and traffic disruptions triggered by natural impacts, which led to emergency situations of
different scale and severity, was  used .
Risk indicators were calculated for each federal region as average annual numbers of
emergency situations in  each type of transport, as
well as a resulting average annual number of emergencies due to all transport accidents and disruptions.
Thus, the calculated indicators included the probability of undesirable consequences (emergencies) due to
impacts of natural hazards on transport infrastructure exposed and vulnerable to these influences.
Quantitative and qualitative criteria for classifying transport accidents and disruptions as emergency
situations are listed above. For the analysis, the period from 1992 to 2018 was chosen, since it covered the
information accumulated in the database.
Additionally, all the federal regions were divided into groups  according to their  risk level.
The risk level was estimated for each federal region and each type of transport by the average annual
number of emergency situations in comparison with the average value of the indicator in Russia. The
number of groups was determined in each case depending on the dispersion of the calculated value." -
(Lines 596-632)

*Line 255 – The database shows for the short period between 2013 and 2018 accidents due to natural*
*hazards, but hazards have long or short return periods; not considering this aspect, as well as*
*vulnerability and exposure means that you are providing a map reflecting the risk, but a map showing*
*recently affected areas. What if a major earthquake in a not so active area strikes an area with no*
*transport accidents in the last 10 years? Your map will tell that the risk in that area is small, not really*
*helping in mitigation efforts.*

The database covers the period from 1992 to 2018. This period is used in the revised version of the analysis for all modes of transport (not only for railway as previously). During this period, events caused by hydro-meteorological and exogenous natural hazards are mainly recorded in the database. Nevertheless, the most seismically active regions of Russia have the highest risk indicator as a result of the assessment. The following explanation was added to the Conclusion section:

"For the study period of 1992 to 2018, the database mainly recorded events caused by exposure to hydro-meteorological and exogenous natural hazards. With high value of the risk index, Kamchatka, Sakhalin, the North Caucasus, and south of Siberia are also among the most seismically active regions of Russia, which further increases the likelihood of emergencies in these regions in case of an earthquake." - (Lines 934-937)

*Line 263 – How is vulnerability considered?*

The vulnerability is considered in the concept of emergency situation, which is used in this study to assess risk. Definition of risk and a detailed description of the method used are included in the methodology section (see response to the comment to line 226). The following explanation was also added to the Conclusion section:

"An annual average frequency of occurrences of emergency situations of various scale and severity severe events was is applied chosen in this study among all possible methods for assessing risk. Unlike methods that assess risk by measuring its components such as hazard, exposure and vulnerability, this approach takes into account the consequences of the above factors and the probability of these consequences. Transport accidents and disruptions are considered in this case as consequences of natural hazard impacts on transport infrastructure that is exposed and vulnerable to these impacts. The risk index is calculated as an annual average number of emergency situations caused by natural hazard impacts in each federal region and each type of transport." - (Lines 912-919)

*Line 266 – Does this correlate with natural hazard maps?*

This does not fully correlate with natural hazard maps. A description of natural hazards in Russia was included in section 2.1:

[revised manuscript text omitted]

*Line 274 – As mentioned before, understanding risk with no consideration of hazard, vulnerability or*
*exposure, but just based on a 5-years statistics window, is certainly not the best instrument to target risk*
*mitigation; especially also since accidents variations are not considerable. Also, the size of the territories*
*is very different – how does this reflect in the analysis?*

Definition of risk and a detailed description of the method used are included in the methodology section
(see above responses to the comments to line 226 and 263).

*Line 279 – Not well referenced.*

The citation of this reference was revised as follows: (Malkhazova and Chalov, 2004). Instead of the title
of the book, the names of the editors were used.

*Line 281 – Can you please provide an evidence?*

The sentence was modified as follows:

"Other factors, such as growing transportation network, increased traffic, and the lack of funding will also
lead to increasing of adverse impacts, especially  with further development of transport
infrastructure to areas with high level of natural  risk." (Lines 944-946)

*Line 298 – Given the potential usefulness of the mentioned database I think that is a limitation not to*
*share this database with the community, also in the purpose of validation and verification.*

The sentence was modified as follows:

"The data used in this study are collected by the author in an electronic database, which is not confidential
and property of Lomonosov Moscow State University and cannot be made available publicly".

*Table 1 - Volcanic eruption - Volcanic eruptions can clearly affect air transport (see what happened in*
*Iceland a couple years ago) and in some cases water transport.*

I absolutely agree with the reviewer that volcanic eruptions can affect air transport. Table 1 reflects only
accidents and disruptions that occurred in Russia. However, the volcanic eruption in Iceland really
affected Russian airports. I added these incidents to Table 1. The following explanation was also included
in section 3.1.3:

"For the study period, there was not a single accident caused by volcanic eruption in Russia. Due to the
eruption of the Icelandic volcano Eyyafyatlayokudl, airlines canceled and delayed more than 500 flights
at 10 Russian airports in April 2010; 32 thousand passengers could not fly." - (Lines 775-777)

*Snow avalanche – Only if the airport is close to the avalanche area probably; in this situation, also water*
*transport could be blocked by rock fall.*

As is indicated in the heading: "Transport accidents and traffic disruptions caused by natural hazards in
Russia (1992-2018)", Table 1 reflects only real accidents that occurred in Russia. The accident on April
10, 2010 in Kamchatka was recorded in the database when a helicopter was damaged as a result of an
avalanche. The explanation was included in section 3.1.3 (Lines 773-774). No cases were recorded in the
database when water transport was blocked by rock fall.

*Figure 2. – It would be interesting to have at least the headers in English, to understand what the*
*database accounts for.*

Figure 2 was replaced by the following description of the database structure in Section 2.2:

"The main database table, into which all the information is entered, has the following structure:

1) event number - the number changes automatically as information is entered;
2) date of the incident;
3) country;
4) region;
5) location - the distance to the nearest settlement is additionally indicated;
6) type of accident - according to the EMERCOM classification and assessment by the author;
7) a brief description of the event, including the time of occurrence, probable cause of the accident,
if available, its consequences, and measures taken to eliminate them;
8) geographical coordinates, if applicable;
9) the scale of the emergency situation caused by the accident – local, inter-municipal, regional,
inter-regional, cross-border;
10) the number of deaths;
11) the number of injuries;
12) economic and environmental losses, if any;
13) source of information." - (Lines 528-544)

*Figure 3. – I would prefer to see the labels (names of regions) in English, in order to identify places*
*mentioned in the text. This applies to all maps.*

A new Figure 2 with names of regions in English was included in the revised version of the manuscript.
All the federal regions, which are mentioned in the manuscript, are indicated in Figure 2.

*Figure 3 – How come there are no values between 2.5 and 3.0 or 4.5 and 5?*

Figure 3 was revised to reflect the new assessment results.

*Figure 5 – How come there are no values between 2.5 and 3.0 or 4.5 and 5?*

Figure 5 was revised to reflect the new assessment results.

*Do the air and water transportation accidents are included in the risk analysis?*

Yes, the air and water transportation accidents are included in the risk analysis. The explanation was
added to section 2.2:

[revised manuscript text omitted]

---

## Author Response (AR2)

I thank the Editor for the very useful comments that allowed me to improve the manuscript. The Editor's comments were taken into account in the revised version of the manuscript, as explained below. The Editor's comments are in italics, the answers are in black and the changes made to the text are in red. The lines numbers refer to the lines numbers of the revised manuscript.

*Comments to the Author:*
*I thank the referees for thourough comments and the author for responding to those comments.*
*I have some observations before the paper can be published:*
*- referee #1 comments on why the proposed classification was preferred to other published*
*schemes. Although the author's response is convincing, I suggest to include in the references some published schemes for comparison.*

Some published classifications of transport infrastructure and natural hazards are included in the introduction and in the references:
"In studies on the impacts of natural hazards, transport infrastructure is most often classified by mode of transport including road, rail, water, and air transport (e.g. Govorushko 2012; Mattsson and Jenelius, 2015; Voumard et al., 2018). Some researchers classify it by infrastructure asset types. For example, Kaundinya et al. (2016) select such transport assets as bridges, tunnels, embankments, cuts and centralized systems. This analysis is structured by mode of
transport." - (Lines 29-33)
"Natural processes and phenomena can be classified in various ways depending on the objectives of a study. Natural hazards can be typify according to their genetic features (e.g. Voumard et al., 2018), the intensity of their manifestation, the main formation and development factors, characteristics of spatial distribution and mode, etc. (Malkhazova and Chalov, 2004). Liu et al.
(2016) propose a systematic natural hazard interaction classification based on the hazard-forming environment. Gill and Malamud (2016) propose a detailed classification of natural-hazard types in Guatemala, including six natural-hazard groups (geophysical, hydrological, shallow Earth processes, atmospheric, biophysical, and space), 19 hazard types, and 37 hazard sub-types." - (Lines 41-48)
Additionally, I revised Figure 1 by replacing it with a better resolution file. – P. 17.

*- line 189-210 (database description). I suggest to include a graphic showing the relational structure of the database. This is very useful in papers dealing with databases.*
A graphic showing the relational structure of the database was included in the manuscript as a
new Figure 3. – P. 19. Accordingly, the numbers of the figures following this Figure were changed to Figures 4-6.

*- the definition of vulnerability and risk as considered in this study should be moved more to the begin of the methodology section.*
Paragraphs containing the definition of vulnerability and risk have been moved to the beginning of the methodology section. – (Lines 193-228)

*- which was the collection method of the author, which were the sources (media, archives, data of municipalities)? This should be briefly mentioned. How difficult are they to be accessed? Who*
*can access them?*
The collection method of the author is described in the methodology section. This description has been revised as follow:
The database is constantly updated with new information. Currently, it contains about 20 thousand events from 1992 to 2018. Official daily emergency reports of the EMERCOM[1] of
* * *
[1] The Ministry of the Russian Federation for Civil Defense, Emergencies and Elimination of Consequences of Natural Disasters

Russia and media reports serve as data sources. Only open data is used. Emergency reports are publicly available on the EMERCOM website (https://www.mchs.gov.ru), but only in Russian. - (Lines 232-235)

*I welcome the author's answer regarding hazard maps, as it was a comment I would have*
*otherwise made.*
Thank you again for your helpful comments.

[revised manuscript text omitted]
 emerg̶e̶n̶c̶y̶ ̶s̶i̶t̶u̶a̶t̶i̶o̶n̶s̶ ̶i̶n̶ ̶e̶a̶c̶h̶ ̶t̶y̶p̶e̶ ̶o̶f̶ ̶t̶r̶a̶n̶s̶p̶o̶r̶t̶,̶ ̶a̶s̶ ̶w̶e̶l̶l̶ ̶a̶s̶ ̶a̶ ̶r̶e̶s̶u̶l̶t̶i̶n̶g̶ ̶a̶v̶e̶r̶a̶g̶e̶ ̶a̶n̶n̶u̶a̶l̶ ̶n̶u̶m̶b̶e̶r̶ ̶o̶f̶
e̶m̶e̶r̶g̶e̶n̶c̶i̶e̶s̶ ̶d̶u̶e̶ ̶t̶o̶ ̶a̶l̶l̶ ̶t̶r̶a̶n̶s̶p̶o̶r̶t̶ ̶a̶c̶c̶i̶d̶e̶n̶t̶s̶ ̶a̶n̶d̶ ̶d̶i̶s̶r̶u̶p̶t̶i̶o̶n̶s̶.̶ ̶T̶h̶u̶s̶,̶ ̶t̶h̶e̶ ̶c̶a̶l̶c̶u̶l̶a̶t̶e̶d̶ ̶i̶n̶d̶i̶c̶a̶t̶o̶r̶s̶ ̶i̶n̶c̶l̶u̶d̶e̶d̶ ̶t̶h̶e̶
p̶r̶o̶b̶a̶b̶i̶l̶i̶t̶y̶ ̶o̶f̶ ̶u̶n̶d̶e̶s̶i̶r̶a̶b̶l̶e̶ ̶c̶o̶n̶s̶e̶q̶u̶e̶n̶c̶e̶s̶ ̶(̶e̶m̶e̶r̶g̶e̶n̶c̶i̶e̶s̶)̶ ̶d̶u̶e̶ ̶t̶o̶ ̶i̶m̶p̶a̶c̶t̶s̶ ̶o̶f̶ ̶n̶a̶t̶u̶r̶a̶l̶ ̶h̶a̶z̶a̶r̶d̶s̶ ̶o̶n̶ ̶t̶r̶a̶n̶s̶p̶o̶r̶t̶
i̶n̶f̶r̶a̶s̶t̶r̶u̶c̶t̶u̶r̶e̶ ̶e̶x̶p̶o̶s̶e̶d̶ ̶a̶n̶d̶ ̶v̶u̶l̶n̶e̶r̶a̶b̶l̶e̶ ̶t̶o̶ ̶t̶h̶e̶s̶e̶ ̶i̶n̶f̶l̶u̶e̶n̶c̶e̶s̶.̶ ̶Q̶u̶a̶n̶t̶i̶t̶a̶t̶i̶v̶e̶ ̶a̶n̶d̶ ̶q̶u̶a̶l̶i̶t̶a̶t̶i̶v̶e̶ ̶c̶r̶i̶t̶e̶r̶i̶a̶ ̶f̶o̶r̶
c̶l̶a̶s̶s̶i̶f̶y̶i̶n̶g̶ ̶t̶r̶a̶n̶s̶p̶o̶r̶t̶ ̶a̶c̶c̶i̶d̶e̶n̶t̶s̶ ̶a̶n̶d̶ ̶d̶i̶s̶r̶u̶p̶t̶i̶o̶n̶s̶ ̶a̶s̶ ̶e̶m̶e̶r̶g̶e̶n̶c̶y̶ ̶s̶i̶t̶u̶a̶t̶i̶o̶n̶s̶ ̶a̶r̶e̶ ̶l̶i̶s̶t̶e̶d̶ ̶a̶b̶o̶v̶e̶.̶ ̶F̶o̶r̶ ̶t̶h̶e̶ ̶a̶n̶a̶l̶y̶s̶i̶s̶,̶
t̶h̶e̶ ̶p̶e̶r̶i̶o̶d̶ ̶f̶r̶o̶m̶ ̶1̶9̶9̶2̶ ̶t̶o̶ ̶2̶0̶1̶8̶ ̶w̶a̶s̶ ̶c̶h̶o̶s̶e̶n̶,̶ ̶s̶i̶n̶c̶e̶ ̶i̶t̶ ̶c̶o̶v̶e̶r̶e̶d̶ ̶t̶h̶e̶ ̶i̶n̶f̶o̶r̶m̶a̶t̶i̶o̶n̶ ̶a̶c̶c̶u̶m̶u̶l̶a̶t̶e̶d̶ ̶i̶n̶ ̶t̶h̶e̶ ̶d̶a̶t̶a̶b̶a̶s̶e̶.̶

[revised manuscript text omitted]

---

## Author Response (AR3)

I thank the Editor for the comments that were taken into account in the revised version of the manuscript, as explained below. The Editor's comments are in italics, the answers are in black and the changes made to the text are in red. The lines numbers refer to the lines numbers of the revised manuscript.

*Comments to the Author:*

*I thank the author for the implemented changes and highlights. I welcome especially the additions on the database structure as table and with the reference to the sources. As a minor correction something on the software for the database could be added.*

I added the software information for the database as follow:

The information collected by the author in an electronic database of technological and natural-technological accidents (created using Microsoft Access) is analyzed in this study. - (Lines 205-206)

*Looking forward to see it published!*

I thank the Editor and both reviewers for their help in improving my manuscript.

[revised manuscript text omitted]